# Quantifying the Ecological Performance of Migratory Bird Conservation: Evidence from Poyang Lake Wetlands in China

**DOI:** 10.3390/biology13100786

**Published:** 2024-09-30

**Authors:** Zhenjiang Song, Shichao Gao, Mingni Leng, Bo Zhou, Baoshu Wu

**Affiliations:** 1Institute of New Rural Development, Jiangxi Agricultural University, Nanchang 330045, China; tgsongzhenjiangl@126.com; 2College of Economics and Management, Jiangxi Agricultural University, Nanchang 330045, China; gaoshichao316@163.com (S.G.); lengmingni@126.com (M.L.); 3School of Business Administration, Jiangxi University of Finance and Economics, Nanchang 330032, China

**Keywords:** avian conservation, coexistence of humans and birds, ecological modeling, ecological network, GIS, habitat quality, Poyang Lake

## Abstract

**Simple Summary:**

Protected areas are essential for the conservation of biodiversity. (i) Ecological connectivity is essential for the maintenance and support of migratory bird habitats. Enhancing control measures to expand ecological corridors can effectively safeguard flagship and umbrella species, thereby promoting biodiversity conservation. (ii) The establishment of ecological corridors can facilitate the reconciliation of conflicts between conservation efforts and development goals, thereby yielding significant theoretical implications for achieving a harmonious coexistence between humans and birds within the migratory bird sanctuary of Poyang Lake.

**Abstract:**

Protected areas are essential for the conservation of biodiversity. However, the rapid expansion of urbanization and the intensification of human activities have significantly disrupted environmental integrity, leading to a continuous deterioration in both the quantity and quality of large ecological patches. This has further diminished the connectivity among ecological patches, leading to significant consequences for regional biodiversity conservation. Taking Poyang Lake as a case study, which serves as a crucial wintering habitat for migratory birds along the East Asia–Australasia flyway, this research employs ArcMap technology. It considers various factors including land use type, slope, and elevation to evaluate habitat quality and degradation through the application of the InVEST model. Additionally, the study utilizes the minimum cumulative resistance (MCR) model alongside circuit theory to delineate ecological corridors within the area and to establish a comprehensive ecological network system. The research results in this paper are as follows. (i) During the period from 2000 to 2020, there was an overall decline in habitat quality within the study area, indicating a clear trend of habitat degradation. However, it is worth noting that there was an increase in habitat quality in certain local areas within the protected area. (ii) The ecological resistance values in the core area of the migratory bird reserve in Poyang Lake are generally low. However, the ecological resistance values of the habitats have shown a consistent increase from 2000 to 2020. Additionally, there has been a significant decrease in the density of ecological corridors during this time period. (iii) Over the period from 2000 to 2020, both the number and connectivity of ecological corridors decreased and their integrity and functionality degraded. Consequently, this weakened role of the ecological network has had implications for maintaining regional biodiversity and ecosystem service functions. The findings indicate two conclusions. (i) Ecological connectivity is essential for the conservation of migratory bird habitats. Strengthening control measures aimed at expanding ecological corridors can effectively safeguard flagship and umbrella species, thereby promoting biodiversity conservation. (ii) The establishment of ecological corridors can help reconcile conflicts between conservation efforts and development objectives. This reconciliation carries significant theoretical implications for fostering a harmonious coexistence between humans and birds in Poyang Lake’s migratory bird sanctuary.

## 1. Introduction

The global phenomenon of climate change is driving a biodiversity crisis, which has elicited widespread international concern owing to the threatened survival of endangered species [1,2,3,4,5]. The habitats of relict species are increasingly shrinking due to the effects of global warming and the rising frequency of extreme weather events. These developments present unprecedented challenges to the survival and growth of these species [6,7]. The climate crisis is exacerbating floods and drought disasters [8,9,10,11], leading to severe degradation or loss of traditional habitats for species [12]. Habitat fragmentation, disruption of waterway connections, and the disappearance of traditional foraging grounds are further worsening the ecological crisis [13]. It is crucial to take urgent measures to enhance the connectivity of wetland ecosystems in order to maintain biodiversity, as it holds significant ecological significance [14,15,16]. Hence, the investigation of ecological connectivity is essential to ensure the habitat security of wetland migratory bird sanctuaries.

Previous research has made considerable progress in the conservation of habitats within designated protected natural areas [17,18]. Research in the domain of habitat ecological connectivity has shown that it effectively mitigates the negative impacts of fragmentation. This enhancement is crucial for ensuring species’ foraging [19], survival, and reproduction [20,21,22], while concurrently optimizing the structure of ecological networks [23,24,25,26]. Alternatively, disturbances induced by human activity may lead to fragmented habitats that obstruct both individual and genetic exchanges, disrupt foraging patterns, and increase the risk of species extinction [27,28]. Urbanization acts as a significant catalyst for the fragmentation and depletion of wetland habitats, consequently hindering ecological connectivity within these environments [29]. The research conducted by Dong et al. [30] has demonstrated that the rapid process of urbanization and changes in land use are leading to significant habitat range erosion. This phenomenon exacerbates alterations in ecological patterns within habitats, contributes to fragmentation issues, diminishes landscape connectivity, and poses a considerable threat to the survival of aquatic and amphibian species due to impeded individual dispersion and genetic exchange. Further investigations conducted in migratory bird sanctuaries have revealed that the decline in species numbers [2,31] and habitat quality [24,32,33,34] can primarily be attributed to the reduction and fragmentation of traditional advantageous habitats. Additionally, the observed decrease in ecological connectivity within these habitats has been shown to significantly affect the integrity of avian food chains [13,35,36], thereby posing a considerable threat to population sizes [37,38,39,40,41]. Furthermore, the primary distinction between migratory birds and mammals is reflected in their migratory behavior. This behavior involves annual movements across various habitats, each of which possesses characteristics typical of stopover sites. The ecological connectivity within the region is crucial for assessing the habitat quality of these stopover sites [42]. However, recent anthropogenic activities have led to the degradation of stopover site habitats, adversely affecting the migratory patterns of avian species [43,44]. This has consequently compromised their survival and reproductive capacities during migration [31,45], posed significant threats to their safety along the migratory route, and precipitated a rapid decline in population sizes [46,47], which may potentially lead to local extinctions [48,49,50].

Currently, the evaluation of habitat quality predominantly emphasizes both species-level [51,52] and ecosystem-level assessments [53,54]. Species-scale assessment, primarily based on fixed-point observations, employs a micro-oriented approach that is particularly effective for examining species evolution. However, it may not support public policy research effectively. Ecosystem-scale investigations integrate macro-level policies, economics, social factors, and ecological considerations to provide a comprehensive assessment of habitat quality for flagship species in specific regions. This approach commonly utilizes comprehensive assessments of environmental services and trade-off models, such as InVEST [24,55,56] and GeoSOS [57], incorporating disciplines like ecology, geography, and economics. For example, Xue et al. evaluated the ecosystem services in the Sanjiangyuan region using the InVEST model and the USLE equation [55,58]. Wei et al. utilized the PLUS-InVEST model to evaluate spatiotemporal changes in habitat quality within the Ebinur Lake watershed [56]. Miguel et al. conducted an assessment of the conservation status of habitats in the Marquesas Islands utilizing the InVEST model. The InVEST model is frequently utilized for evaluating habitat quality in relation to land use changes [57]. In wetland systems, the minimum cumulative resistance model is employed to develop an ecological resistance surface. This approach facilitates the establishment of ecological networks grounded in circuit theory, thereby enabling the assessment of habitat quality within migratory bird sanctuaries [54].

This study aims to investigate the temporal changes in the Poyang Lake migratory bird habitat conservation area from 2000 to 2020 by examining relevant literature, while also assessing the efficacy of conservation policies. This study investigates the spatiotemporal dynamics of habitat quality within the Poyang Lake Migratory Bird Protection Area, utilizing the InVEST model [24,57,59], minimum cumulative resistance model [60], and circuit theory [61], which were employed to clarify the spatial patterns of land use for migratory bird habitats at Poyang Lake. These findings provide a significant reference for advancing the exploration of policy mechanisms that foster harmonious coexistence between humans and birds. The primary objectives of this study are to evaluate and visually represent the spatiotemporal distribution of habitat quality for migratory birds in Poyang Lake from 2000 to 2020. Additionally, the study aims to analyze the influence of China’s biodiversity conservation policies on enhancing migratory bird habitat quality, as well as the corresponding trends over this two-decade period. This paper consists of five sections. The second section provides an overview of the study area, data sources, and research methods. The third section examines the changes in habitat quality in the study area from 2000 to 2020, develops an ecological resistance surface, identifies ecological corridors, and establishes an ecological network for analyzing the pattern of ecological change. The fourth section presents the research conclusion and policy implications. The fifth section includes further discussion on the topic.

## 2. Materials and Methods

### 2.1. Study Area

Poyang Lake, the largest freshwater lake in China and a significant tributary of the middle and lower reaches of the Yangtze River, undergoes seasonal regulation to form a wetland ecosystem that serves as an exceptional wintering habitat for migratory birds from northern regions (Figure 1). To protect rare migratory bird species such as the Siberian crane and their wintering areas, a nature reserve was established in 1983 and subsequently elevated to national nature reserve status in 1988. This reserve provides protection for an extensive range of biodiversity including 310 avian species, 122 fish species, 15 mollusk species, 46 plankton species, 227 insect species, 13 amphibian species, 50 reptile species, and 31 mammal species. Moreover, it plays a crucial role in water regulation dynamics and climate moderation processes while also functioning as a vital habitat for wildlife.

Despite the gradual enhancement of China’s natural resource protection system in recent years, the issue of “human–bird land competition” at the edges of wetlands continues to encroach upon bird habitats. This situation highlights a persistent contradiction between conservation and development. As a consequence of anthropogenic disturbances, including land encroachment around the lake for agricultural activities, the habitats of migratory birds have experienced considerable shrinkage. This phenomenon has given rise to a complex interface referred to as the human–bird coexistence system [62,63]. The coexistence system examined in this study illustrates the simultaneous manifestation of bird damage and human disturbances. Human activities, including agriculture, tourism, industrial development, and urban expansion, have profoundly disrupted the wetland habitats that serve as crucial environments for waterfowl. This disruption has led to a persistent decline in migratory bird habitats and an increase in conflicts between humans and birds [64,65]. This presents a significant threat to the survival of migratory bird species, with some populations categorized as endangered or vulnerable [66,67]. Meanwhile, over the past four decades, concerted efforts in bird conservation have consistently enhanced the population structure of migratory birds. However, this growth has led to a paradoxical situation in which the available habitat area is unable to keep pace with the expanding population due to human disturbances and encroachment. Consequently, essential habitats vital for migratory birds have been diminished, forcing these species to seek nourishment in areas impacted by human activities. During the late maturation of rice, migrating birds that travel southward often feed on rice grains, which leads to detrimental effects on human livelihoods and substantial grain losses [68]. Given the inclusion of migratory birds in the protective legislation and the prohibition of hunting within the regulatory system, effectively mitigating bird damage issues poses a challenge for human populations. This ongoing conflict obstructs the harmonious coexistence of humans and birds, consequently hindering regional sustainable development. However, wetlands function not only as habitats for avian species but also as the foundations of human civilization and crucial centers for the production of commodity grains [69,70]. It is impractical to resolve ecological conflicts through a step forward, step back approach. Therefore, to effectively mitigate human–bird conflicts, it is crucial to undertake a comprehensive assessment of migratory bird habitats and analyze the specific characteristics associated with these conflicts between humans and birds. This will serve as a valuable reference for enhancing institutional frameworks and designing effective mechanisms. Therefore, this study focuses on the Poyang Lake Migratory Bird protection Area as the research site, which includes the nature reserve, designated conservation areas, adjacent communities, and their corresponding counties. It evaluates both the quality of habitat within these areas and the broader potential range for these habitats. This approach provides valuable data and mechanisms to support the conservation of migratory bird habitats, promote biodiversity preservation, and foster a harmonious coexistence between humans and nature.

### 2.2. Data Sources

#### 2.2.1. Map Data

The distribution map data for the Poyang Lake Migratory Bird Protection Area is derived from the inspection report compiled by the Forestry Department of Jiangxi Province. The administrative division base map of the sanctuary is sourced from the 1:1,000,000 National Basic Geographic Database (https://www.webmap.cn/, accessed on 1 August 2024). These maps utilize the 2000 National Geodetic Coordinate System and the 1985 National Elevation Datum, with data segmented into sections using geographic coordinates in degrees as units of measurement.

#### 2.2.2. Land Use Data

The land use data utilized in this study encompass two time periods, namely, 2000 and 2020. The remote sensing images for these datasets were obtained from the Landsat TM/ETM+/OLI images publicly accessible through the Geospatial Data Cloud of the Computer Network Information Center at the Chinese Academy of Sciences (https://www.gscloud.cn/, accessed on 1 August 2024). These images exhibit consistent spectral bands, strip numbers, and row numbers, with imaging conducted between June and September of each respective year to ensure minimal cloud cover and facilitate accurate identification of ground objects. Following processing using ENVI 5.3 software, the image coordinates were standardized to the GCS_WGS_1984 reference system. The support vector machine classification (SVM) method was employed along with interactive human–computer interpretation to extract original spatial distribution maps depicting land use for both time periods [71,72]. Subsequently, post-classification processes including clustering and filtering techniques were applied to derive land use/cover change (LUCC) data for both time periods, whose resolution is 1000×1000 (Figure A1).

#### 2.2.3. Migratory Bird Footprint Line Observation Data

Starting from October 2020, annually between 1 October and 30 November, a network of 48 specialized industry bases was established around the Poyang Lake Migratory Bird Protection Area community. These bases were organized under the auspices of the National New Rural Development Research Institute of Jiangxi Agricultural University (sponsored by the Ministry of Science and Technology and the Ministry of Education). The purpose was to set up migratory bird footprint sampling lines, with a focus on the flagship species, the Siberian crane. Each base was equipped with 15 sampling lines, resulting in a total of 720 observation points. These points encompassed various data including encounters with migratory birds, water levels, soil moisture content, accumulated temperature, local household numbers, as well as average farmer income. To reconstruct migratory bird footprint data for 2000 accurately and reliably, an exponential smoothing method was employed based on scientific expedition reports conducted at Poyang Lake from 2010 to 2019.

### 2.3. Analytical Methods

The study employed ArcMap 10.8 software [73,74], in conjunction with the InVEST model, to compute the outcomes of habitat degradation and habitat quality. Subsequently, the minimum cumulative resistance model (MCR) was utilized to construct an ecological resistance surface [54], and by employing circuit theory [75], ecological corridors were identified [75], systematically evaluating the level of habitat security within the Poyang Lake Migratory Bird Protection Area.

#### 2.3.1. InVEST Habitat Quality Model

The InVEST (integrated valuation of ecosystem services and tradeoffs) model, jointly developed by Stanford University, the World Wildlife Fund, and The Nature Conservancy, serves as a valuable tool for assessing ecosystem services and analyzing habitat quality. The quality of the ecological environment is assessed and quantified by analyzing the stress impacts exerted by various factors on the ecosystem. The ecological role of an ecological environment is positively correlated with its species richness [59]. In this study, the habitat quality module in the InVEST 3.14.0 model was employed to assess the habitat quality of the Poyang Lake Migratory Bird Protection Area. The calculation formula proposed by Chen et al. [59] was utilized for this purpose.
(1)$$Q_{xj}=H_{xj}\left(1-\frac{D_{xj}^{z}}{D_{xj}^{z}+k^{2}}\right),$$
(2)$$D_{xj}=\sum_{r=1}^{R}\sum_{y=1}^{Yr}\left(\frac{w_{r}}{\sum_{r=1}^{R}w_{r}}\right)r_{y}i_{rxy}\beta_{x}S_{jr}$$

In the equation, *Q_xj_* represents the habitat quality of grid *x* in land use type *j*. *H_xj_* denotes the habitat suitability of grid *x* in land type *j*. *D_xj_* quantifies the degree of habitat degradation for grid *x* in land type *j*. *k* is defined as the semi-saturation constant, which is set to half of the maximum value representing habitat degradation after running the model once. *z* serves as a normalization constant, with a default value of 2.5 [76]. The habitat quality value ranges from 0 to 1, with a higher *Q_xj_* value indicating superior habitat quality. *Y_r_* is the number of threat factor rasters; *w_r_* is the proportion of threat factors. *R* is the number of threat factors; *r_y_*, *i_rxy_*, *β_x_* and *S_jr_* were the intensity of threat factors, the influence on habitat quality, the influence on policy, and the sensitivity of land class *j* to stress factor *r*, respectively. Considering the existing landscape pattern and previous research conducted in the Poyang Lake Migratory Bird Protection Area [59,76,77], combined with the field situation in the study area, threat factors were identified as arable land, urban areas, rural settlements, and other construction lands that experience significant disturbances caused by human activities. The distance algorithm between habitat rasters and threat is shown in Formulas (3) and (4) [78]:(3)$$i_{rxy}=1-\left(\frac{d_{xy}}{d_{r}\max}\right),\mathrm{linear}$$
(4)$$i_{rxy}=\exp\left(-\left(\frac{2.99}{d_{r}\max}\right)d_{xy}\right),\mathrm{exponential}$$

The maximum threat distance, relative weight, and spatial decay type for each threat factor were sequentially set according to existing research (Table 1). Values were assigned for the habitat suitability of different land use types and their sensitivity to threat factors (Table 2).

#### 2.3.2. Construction of Ecological Resistance Surface

Ecological Resistance Assessment Indicators.

The establishment of an ecological resistance surface is a crucial prerequisite for the delineation of ecological corridors, as it reflects the impediments encountered by ecological flows during energy transfer, material exchange, or species migration between ecological patches [79]. Based on a comprehensive review of pertinent studies and taking into account the specific conditions of the Poyang Lake Nature Protection Area, four ecological resistance factors have been meticulously selected for the Poyang Lake Migratory Bird Protection Area. These factors are land use type, digital elevation model (DEM), normalized difference vegetation index (NDVI) (Figure 2), and a slope distribution map (Table 3) (Figure 3) [80]. Utilizing the analytic hierarchy process (AHP), the resistance indicator factors were systematically categorized into five hierarchical levels, wherein higher values signify increased resistance coefficients.

Minimum Cumulative Resistance (MCR) Model.

The widely adopted minimum cumulative resistance (MCR) model is employed in this article to construct a comprehensive resistance surface. The formula proposed by Zhang et al. [56] is utilized for this purpose.
(5)$$MCR=fmin\sum_{j=n}^{i=m}\left(D_{ij}R_{i}\right)$$

In the formula, *MCR* represents the minimum cumulative resistance value for a specific ecological source patch in the region to spatially propagate to a particular point. *f* denotes the positive correlation between minimum cumulative resistance and ecological processes, *D_ij_* signifies the spatial distance from source *j* to landscape unit *i*, and *R_i_* indicates the resistance coefficient of landscape unit *i* towards species movement. The ecological resistance assessment indicators are determined based on existing research, and the study area’s ecological resistance surface is constructed using ArcMap 10.8 software.

Morphological Spatial Pattern Analysis (MSPA).

Morphological spatial pattern analysis (MSPA) pertains to the identification, processing, and classification of land use data in spatial morphology for the acquisition of patch types that hold significance in maintaining landscape connectivity [81]. This paper adopts the land use status data of 2000 and 2020 as the foundation and makes use of the classification tool in ArcGIS 10.8 to classify forests, grasslands, water bodies, croplands, and wetlands as the analytical foreground data and assign them a value of 2. Meanwhile, it classifies urban and rural settlements as the background data and assigns them a value of 1. In the event of any missing data, the byte is set to 0, thereby creating a binary raster map. By utilizing Guidos Toolbox 2.8 software for MSPA, a total of 7 distinct landscape types were identified (core area, island patches, pores, edge zone, ring island zone, bridge zone, and feeder zone), and the area and the number of patches from the analysis results were subjected to statistical analysis. The quantity of core patches initially decreases and subsequently stabilizes as the area expands. Once the patch area attains 20 km^2^, the number of patches exhibits a gentle trend. Therefore, a minimum area threshold of 20 km^2^ was established to eliminate fragmented and scattered patches, and areas having core areas larger than 20 km^2^ were extracted as the source data for landscape connectivity index analysis, ultimately obtaining ecological source areas for 2000 and 2020.

#### 2.3.3. Construction of Ecological Network Based on Circuit Theory

Circuit Theory

The concept of circuit theory was advanced by McRae et al. (2008) [61], which models the interconnections among diverse ecological landscapes by capitalizing on the random walk characteristic of charges. This model fuses circuit theory with movement ecology by considering ecological landscapes as conductive surfaces and organisms as mobile electrons. The movement of organisms is regarded as current, and a random walk model is erected on this foundation to acquire the path with the least energy expenditure for species migration, namely, ecological corridors.

Extraction of ecological corridor

Ecological corridors serve as significant conduits for the material cycling, gene flow, and information transmission within ecosystems, exerting a vital role in connecting various ecological source areas [20,81]. The construction of ecological corridors presented in this paper is founded on circuit theory [82], taking ecological source areas and ecological resistance surfaces as input elements and being implemented through the Linkage Mapper tool module of ArcMap 10.8 software. The connection degree model grounded on circuit theory considers the traits of species’ random migration and is capable of accurately simulating the actual circumstances of species migration [80], thereby constructing an ecological network for the protection of migratory birds in Poyang Lake.

Identification of Ecological Pinch Points and Ecological Barrier Points

Ecological pinch points represent crucial junctions that ensure the connectivity among sources and exert an irreplaceable role in maintaining the integrity of ecosystems [83]. Due to the irreplaceable characteristic of ecological pinch points, organisms are prone to select these areas during their migration. Consequently, any damage to these areas would exert a fatal influence on the integrity of the ecosystem and biodiversity. Hence, the ecological pinch point areas constitute a key protected area within the entire study region. Utilizing the Pinchpoint Mapper tool within the Linkage Mapper module, which integrates minimum cost path and circuit theory in the study area, Circuitscape v4.0.5 software was initially employed to identify areas with high current density. Subsequently, Pinchpoint Mapper was utilized to eliminate areas that contribute minimally to regional connectivity, thus identifying the path of least resistance and critical pinch points. This ultimately leads to the identification of key node areas within the ecological corridor of the study area. This paper identified ecological pinch points in the Poyang Lake Wetland Bird Reserve using the raster centrality option in the all-to-one pattern recognition model. The all-to-one pattern establishes a connection between a core area and the ground, and then distributes current into the remaining cores by iterating through all the core areas. Based on the iterative results, the ecological pinch points in the Poyang Lake Wetland Bird Reserve were determined.

Ecological barrier points refer to areas that hinder the quality of ecological corridors between source areas [29]. The occurrence of barrier points is influenced by multiple factors, and is typically more significantly affected by human interference. Based on the land use types in the study area, distinct solutions are adopted for various problems to minimize the resistance value in the region and enhance the connectivity among landscapes as much as possible through ecological restoration. In this study, the barrier mapper tool within the linkage mapper module was employed to identify highly obstructive areas within the ecological corridor. The minimum search radius was set at 300 m, the maximum search radius at 900 m, and the step size at 300 m. The obstacle points of the study area were searched using the moving window approach. The operation of the barrier mapper tool is based on the linkage pathways tool, which can detect both complete barriers that completely impede species migration and barrier areas that have some degree of impeding effect but are not completely impeding species migration [84]. In this study, only barrier areas that completely impede species migration were considered to determine the ecological barrier point regions in the Poyang Lake Bird Protection Area.

## 3. Results

### 3.1. Spatiotemporal Changes in Habitat Quality

The overall habitat quality of the Poyang Lake Migratory Bird Protection Area demonstrated a positive trend on a spatial scale from 2000 to 2020. Within the defined boundaries of the nature reserve, the quality of habitat was observed to be significantly higher. Conversely, beyond these boundaries, there was a discernible trend indicating a decline in habitat quality due to urban expansion and rural development in the adjacent areas (Figure 2). The extent of habitat degradation was significantly lower within the confines of the nature reserve when contrasted with the adjacent areas. The primary contributors to habitat degradation were identified as urban development and rural settlements. This distribution pattern surrounding the lake is depicted in Figure 3. The central core area of Poyang Lake, which includes the Poyang Lake National Nature Reserve, Duchang Migratory Bird Nature Reserve, Baishazhou Nature Reserve, Kangshan Migratory Bird Nature Reserve, and the Three Lakes Nature Reserve, displayed enhanced habitat quality. In the southern region, natural reserves centered around Junshan Lake and Qinglan Lake demonstrated a notable enhancement in migratory bird habitat within the designated boundaries.

The abundance of high-quality habitat patches within the Poyang Lake Nature Protection Area exhibited a significant increase from 2000 to 2020. The central region of the middle lake area exhibited significant improvements in habitat, particularly with regard to improvements in quality. Moreover, notable enhancements were observed in the habitat quality of Junshan Lake and Qinglan Lake in the southern lake areas. However, the development of the urban circle surrounding Poyang Lake has led to a significant deterioration in habitat quality for migratory bird sanctuaries, particularly in areas marked by high human activity intensity. This is exemplified by the built-up area of Nanchang city, which continues to grow and exacerbate these challenges. The enhancement in habitat quality within the designated boundaries of the nature reserve is markedly greater than that observed in areas beyond these limits.

The temporal scale revealed a normal distribution of habitat quality in both years (Figure 2). The recent enhancement of regulatory control over nature reserves has led to a significant enhancement in the ecological performance levels within the regulated boundaries. This development elucidates the phenomenon being observed. Poyang Lake, serving as a key hub within the Yangtze River Economic Belt and an essential grain production base, has witnessed significant advancements over the past two decades. These developments have occurred within the strategic framework of the construction of the Yangtze River Economic Belt. The expansion of construction land has led to its encroachment upon arable land and other secondary habitats, progressively deteriorating the environmental quality of urban wetland habitats. Nonetheless, the predominant trend in habitat optimization for migratory birds persists, effectively enhancing biodiversity.

In terms of habitat degradation, a declining trend is evident (Figure 3). Over the past two decades, the Chinese government has expedited the implementation of nature conservation initiatives by establishing a comprehensive network of protected areas centered around the Poyang Lake National Nature Reserve. Additionally, ecological protection red lines and urban development boundaries have been delineated to regulate human activities encroaching upon nature reserves. Moreover, China has implemented stringent wildlife protection legislation, with the criminalization of avian hunting deeply ingrained in the public consciousness. Furthermore, numerous non-governmental organizations, such as WWF, have actively undertaken conservation initiatives, effectively mitigating ecological degradation concerns within designated nature reserves. However, the construction of the urban belt encircling Poyang Lake with Nanchang at its core poses an inevitable threat to migratory bird habitats, leading to the transformation of areas such as Yaohu Nature Reserve into urbanized lakes. The issue of “human–bird land competition” beyond the designated boundaries of nature reserves remains a pressing concern.

The habitat quality of the Poyang Lake Migratory Bird Protection Area exhibited a normal distribution pattern in both 2000 and 2020, with habitat levels predominantly concentrated within the range of (0.1, 0.3]. The overall level of habitat quality in 2020 surpassed that of 2000. However, when considering middle and low levels of habitat quality, the year 2000 exhibited higher values compared to 2020 within the ranges (0.05, 0.1], (0.1, 0.3], and (0.3, 0.5].

The extent of habitat degradation within the red boundaries of the nature reserve witnessed a significant increase in 2020, as compared to that in 2000. The extent of habitat degradation in urban built-up areas outside the boundaries of the nature reserve has progressively intensified. Over a span of 20 years, there has been a substantial increase in the density of habitat degradation surrounding the lake.

In both 2000 and 2020, the degree of habitat degradation in the Poyang Lake Migratory Bird Protection Area exhibited a decreasing trend, with the concentration of habitat degradation falling within the range of (0, 0.015]. However, there was a significant increase in habitat degradation in 2020 compared to that in 2000. Specifically, within the interval (0, 0.05], the level of degradation was significantly higher in 2000 than in 2020; whereas, within the interval (0.05, 0.4], the level of degradation was significantly higher in 2020 than in 2000.

### 3.2. Formation of Ecological Resistance Surface and Identification of Ecological Corridors

The ecological resistance surface (Figure 4) indicates that the Poyang Lake Migratory Bird Protection Area experienced a general trend of low ecological resistance values in its core area (within the nature reserve boundaries) and high habitat resistance values due to the expansion of surrounding urban built-up areas and rural settlements during the period from 2000 to 2020. The Poyang Lake Migratory Bird Protection Area (Figure 4) exhibited a peak comprehensive ecological resistance value of 186.75 in the year 2000, with an average ecological resistance value of 2.32. In 2020, the highest recorded ecological resistance value reached 91.3, whereas the average ecological resistance value stood at 2.48. Despite the expansion of human activity areas in the surrounding cities over a span of two decades, there was only a marginal change observed in the average ecological resistance value. During both time periods, the obstructive capacity is distributed in accordance with land use types, exhibiting greater resistance in surrounding urban areas and rural settlements while displaying lower and more consistent resistance within the nature reserve. This factor plays a pivotal role in determining the distribution and orientation of ecological corridors, ultimately resulting in their dense overall concentration centered around Poyang Lake’s core area.

Regarding spatial distribution, ecological source areas are predominantly concentrated in the central Poyang Lake basin. The land use types within the ecological source area region are mainly water bodies, followed by forests, and there are also some unused land and cropland distributed therein, but with a relatively small area. Based on MSPA analysis, seven types of ecological landscapes in 2 years were obtained (Figure 5).

By establishing ecological corridors surrounding the Poyang Lake Migratory Bird Protection Area (Figure 6) and conducting an analysis of ecological pinch points and barrier points (Figure 7 and Figure 8), this study has unveiled a significant decline in ecological corridor density from 2000 to 2020, highlighting the substantial adverse impact of urbanization on such densities. The central Poyang Lake Nature Reserve exhibits high habitat quality, characterized by low levels of habitat degradation and ecological resistance surfaces, thus providing suitable habitats for migratory birds. However, human activities in the past two decades have resulted in the degradation of short-distance ecological corridors that connect adjacent patches, thereby impeding inter-patch species exchange and development. This has significantly constrained species interaction between surrounding woodland areas and the central region of Poyang Lake. These ecological corridors, predominantly reliant on linear features such as rivers, lakes, and forested areas, constitute a centripetal network centered around the Poyang Lake region, exemplifying a distinct centripetal distribution. The corridor distribution during both periods exhibits a dense arrangement surrounding the central Poyang Lake area, indicating that human-activity-intensive urban zones have not severed connectivity between corridors and the overall network. The density of corridors has significantly decreased over the past 20 years, resulting in a pronounced degradation of habitat quality in migratory bird habitats by 2020 compared to 2000.

In this paper, ecological pinch points in the Poyang Lake bird protection area are identified based on the all-to-one pattern recognition model. The cumulative current values among each ecological source area were computed, and the natural breakpoint method was employed to categorize them into three types, designated as Level 1–3 pinch points. The ecological pinch point with the highest current value is recognized as the ecological pinch point (Figure 2 and Figure 5). The identified areas of ecological pinch points in 2000 and 2020 were 98.54 km^2^ and 55.45 km^2^, respectively, presenting a general trend of contraction. Regarding spatial distribution, ecological pinch points are typically situated at the two ends of ecological corridors adjacent to ecological source areas, which fall within the radiation scope of ecological source areas and can offer substantial support for the migration of organisms. They are mainly distributed in Poyang County, Chaisang District, De’an County, and Dongxiang District. Overall, the ecological pinch points situated in Poyang County and Chaisang District remained relatively stable from 2000 to 2020 on account of the stability of their ecological source areas, functioning as a requisite passage for species migration between the ecological source areas of Poyang Lake and its surrounding ones. The ecological pinch points located in Dongxiang District and Nanchang County witnessed a decrease in size, mainly attributed to human activities in recent years. Additionally, the ecological pinch points in Yongxiu County have also been impacted by human activities, with the affected area gradually shrinking. It is necessary to undertake ecological restoration to sustain the stability of the ecological pinch points to guarantee normal species migration. Meanwhile, the migration route of species in Poyang County has augmented due to the increment in ecological source areas and ecological corridors, forming new small ecological pinch points.

The current recovery values for each ecological source area are calculated and classified into four categories by means of the natural breakpoint method, being defined as level one to four impedance zones. The area with the highest current recovery value is regarded as the ecological impedance point (Figure 2 and Figure 6). The ecological barrier points identified in 2000 and 2020 boasted areas of 200.68 km^2^ and 112.52 km^2^, respectively. The area of the barrier points has manifested a trend of gradual decrease over time. In terms of spatial distribution, ecological barrier points are typically situated in the midst of ecological corridors, receiving scarce ecological radiation from source areas featuring good habitats. Meanwhile, the ecological foundation in the northwest region is frail, and the relationship between humans and nature is uncoordinated and accompanied by notable environmental issues. This has augmented the difficulty of material exchange and information transmission. In 2000, the primary ecological barriers were situated in Nanchang County, De’an County, and Chaisang District. By 2020, the barriers had exhibited a more uniform distribution, and the discrepancies in resistance to species migration among different corridors had gradually diminished. Through the combined analysis of ecological pinch points and ecological barriers, it was discovered that there existed overlapping areas of both, signifying that there were regions with more substantial obstacles along the species’ migration routes. Consequently, priority ought to be accorded to ecological restoration in these areas to guarantee the unobstructed passage of species migration routes.

Through a comprehensive analysis of the spatial patterns and densities of ecological corridors surrounding the Poyang Lake Migratory Bird Protection Area for 2000 and 2020, this study has revealed that ecological corridors exhibited a network distribution that gradually attenuates from the central region of Poyang Lake towards its periphery, effectively connecting dispersed ecological redline areas and highlighting the pivotal role played by Poyang Lake. Long-distance ecological corridors primarily extend from the central Poyang Lake area to surrounding rivers, lakes, and woodlands, underscoring the pivotal role of ecological corridors in upholding biodiversity and ecosystem services.

### 3.3. Construction and Quality Analysis of the Ecological Network

Taking into consideration the habitat quality of the Poyang Lake Migratory Bird Protection Area in 2000 and 2020, in conjunction with the spatial distribution of ecological corridors, this study accentuates the considerable adverse impact of the sparsification of ecological corridors on the ecological network system during the past two decades. Despite functioning as a key hub for the conservation of migratory birds, the core area of Poyang Lake presents abundant water sources, species resources, and extensive wetlands that provide vital stepping stones, resting points, and high-quality food resources with a significant aggregation effect. However, the reduction in and degradation of ecological corridors have compromised the connectivity function of this central area. The Poyang Lake and its surrounding nature reserves incorporate the majority of the aquatic ecosystems within the study area, which are vital for ensuring the quality of stopover sites during bird migration. Nevertheless, the reduction in both the quantity and connectivity of long-distance ecological corridors, coupled with the degradation of corridor integrity and functionality, has significantly impeded the central area’s role in maintaining regional biodiversity and ecosystem service functions as an integral part of a multi-corridor protected area ecological network system (Figure 6 and Figure 7).

The results of this study emphasize the urgent necessity to strengthen the protection and restoration of ecological corridors so as to alleviate further fragmentation of the ecological network and thereby guarantee the conservation of habitat quality within Poyang Lake and its adjacent nature reserves. This imperative measure will effectively support the migratory patterns and habitation demands of avian species, ultimately protecting biodiversity and promoting ecosystem health and stability.

## 4. Discussion

With the continuous intensification of human economic activities, the construction of ecological corridors aimed at optimizing the ecological network system has emerged as a crucial measure to promote human–bird symbiosis within nature reserves and enhance biodiversity. In this paper, we utilized the InVEST model and MCR model in conjunction with circuit theory to construct ecological corridors. We investigated the relationship between ecological connectivity and biodiversity enhancement in the migratory bird reserve of Poyang Lake, revealing the key role of ecological corridors in protecting flagship species. The research in this paper further confirms that human activities consistently disrupt habitat connectivity [27,28,29,30], leading to a continuous decline in both biodiversity and habitat quality [2,31,32,33,34]. Furthermore, the decrease in ecological habitat connectivity has already become a threat to migratory bird populations [13,35,36,37,38,39,40,41]. The ecological network constructed in this paper demonstrates the significance of ecological connectivity in Poyang Lake and emphasizes the necessity of increasing the scale of ecological corridors to protect endangered species and their companion species. The construction of ecological corridors can alleviate, to a certain extent, the contradiction in the human–bird symbiosis in the protected area of Poyang Lake and provides a theoretical basis for regional ecological management. However, despite significant progress in methodology, there are still some deficiencies and areas that need further expansion and improvement. Future studies should consider the disturbance factors of human activities, especially the destruction of ecological corridors by urban expansion, in order to comprehensively assess the sustainability of the ecological network. Additionally, simulating the ecological network of Poyang Lake with climate models will help predict and respond to possible environmental changes.

In summary, this paper highlights the significance of ecological corridors in protecting regional biodiversity through the construction of an ecological network in the Poyang Lake region. It also proposes strategies for optimizing the ecological network to improve the regional ecological environment. However, future research should focus on exploring the dynamic changes in the ecological network and its management strategies using more extensive data and multidisciplinary perspectives. This is essential to achieve a balance between ecological protection and regional development, with a goal of realizing human–bird symbiosis.

## 5. Conclusions and Policy Implications

### 5.1. Conclusions

Drawing upon the assessment of habitat quality, construction of ecological resistance surfaces, and establishment of ecological corridors, this study elucidated the intricate network structure of the Poyang Lake Migratory Bird Protection Area to comprehensively investigate the sanctuary’s habitat conditions. The research results in this paper are as follows. ① During the period from 2000 to 2020, there was an overall decline in habitat quality within the study area, indicating a clear trend of habitat degradation. However, it is worth noting that there was an increase in habitat quality in certain local areas within the protected area. ② Ecological resistance values in the core area of the migratory bird reserve in Poyang Lake are generally low. However, the ecological resistance values of the habitats have shown a consistent increase from 2000 to 2020. Additionally, there has been a significant decrease in the density of ecological corridors during this time period. ③ Over the period from 2000 to 2020, both the number and connectivity of ecological corridors decreased and their integrity and functionality degraded. Consequently, this weakened role of the ecological network has had implications on maintaining regional biodiversity and ecosystem service functions. The research presents the following research conclusions.

(1) Ecological connectivity plays a decisive role in shaping the habitat of migratory birds. The enhancement and expansion of ecological corridors are indispensable for the conservation of key and umbrella species, making a considerable contribution to the enhancement of biodiversity. Despite the fact that the ecological network of the Poyang Lake Migratory Bird Reserve encompasses diverse land use types including rivers, lakes, reservoirs, farmland, and forests, and links 14 nature reserves and surrounding cities, the density of ecological corridors has significantly decreased. The reduction has led to a decrease in the protective capacity for associated species, thereby impeding the effective promotion of biodiversity enhancement.

(2) The establishment of ecological corridors plays a vital role in reconciling the conflict between conservation and development, thereby holding significant theoretical and practical implications for achieving harmonious coexistence between humans and birds within the Poyang Lake Migratory Bird Protection Area. The components of the ecological network, such as corridors, stepping stones, and resting points, demonstrate a primary concentration in the central region surrounding the lake. However, during the past two decades, a decrease in the density of ecological corridors has occurred, especially in urban areas characterized by intense human activity, leading to disruptions in connectivity. Nonetheless, the non-existence of this impact is not readily apparent within the natural reserve areas due to effective government management and protection measures. The Poyang Lake Migratory Bird Protection Area serves as a key stopover site for migratory birds, playing an indispensable role in the intricate ecological network system. The multi-corridor ecological network structure centered around Poyang Lake not only tackles the reduction in and fragmentation of ecological patches induced by urbanization but also significantly enhances corridor continuity by means of the construction of long-distance high-quality corridors and the provision of temporary resting spots. The optimization of the ecological network enhances ecosystem resilience, thereby playing a vital role in improving the regional ecological environment and conserving biodiversity. This highlights the strategic importance of concentrating conservation and construction efforts in the areas surrounding the Poyang Lake natural reserve.

### 5.2. Policy Implications

(1) Systematically formulate strategies for the planning of the Poyang Lake Migratory Bird Protection Area, with a primary focus on optimizing the arrangement of ecological corridors, reducing anthropogenic disturbances, and facilitating genetic exchange to enhance biodiversity levels. Develop a comprehensive provincial plan specifically aimed at migratory bird sanctuaries, integrating a dedicated chapter on ecological corridors to establish clear boundaries for corridor protection. This will effectively alleviate disturbances caused by human activities in migratory bird habitats, facilitating genetic exchange and population expansion of both migratory birds and their primary prey fish species.

(2) Facilitate the establishment of ecological corridors through the integration of production, living, and ecological spaces, while redefining the “Two Mountains” concept to alleviate human–wildlife conflicts and foster synergies between conservation and development. Reasonably plan the integration of community farmers and city industrial systems, while meticulously considering ecological corridors. Promote bird-watching and other forms of ecotourism activities to facilitate the transformation of ecological resources into valuable assets. Aim to convert green water and green mountains into sustainable economic benefits. Reformulate protection projects from a developmental perspective to concurrently foster regional economic growth and invigorate conservation efforts.

## Figures and Tables

**Figure 1 biology-13-00786-f001:**
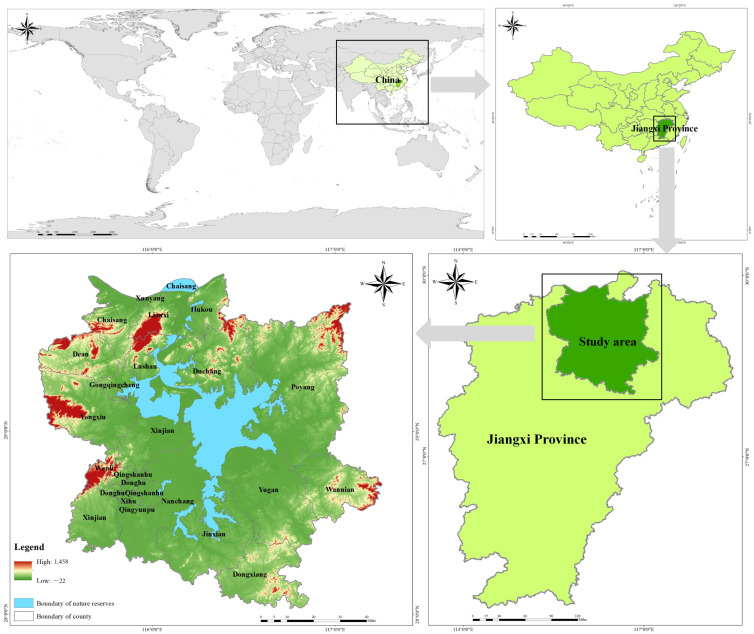
Study area.

**Figure 2 biology-13-00786-f002:**
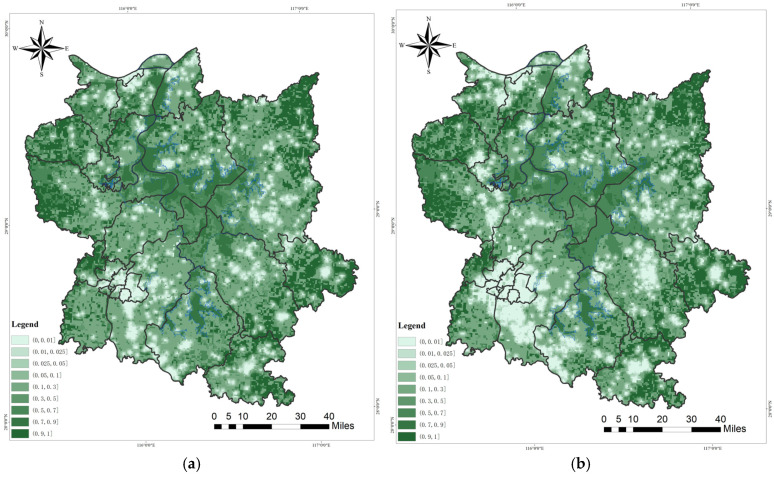
Habitat quality of Poyang Lake Migratory Bird Protect Area: (**a**) 2000; (**b**) 2020; (**c**) habitat quality pattern of Poyang Lake Migratory Bird Protect Area in 2000 and 2020.

**Figure 3 biology-13-00786-f003:**
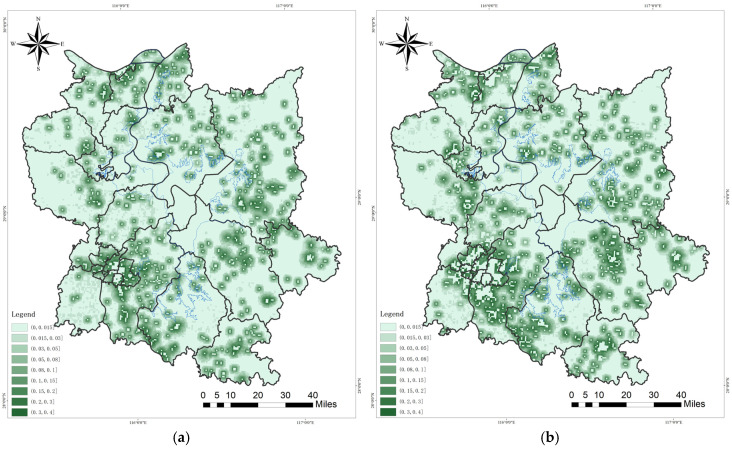
Degree of habitat degradation at the Poyang Lake Migratory Bird Protect Area: (**a**) 2000; (**b**) 2020; (**c**) habitat quality pattern of Poyang Lake Migratory Bird Protect Area in 2000 and 2020.

**Figure 4 biology-13-00786-f004:**
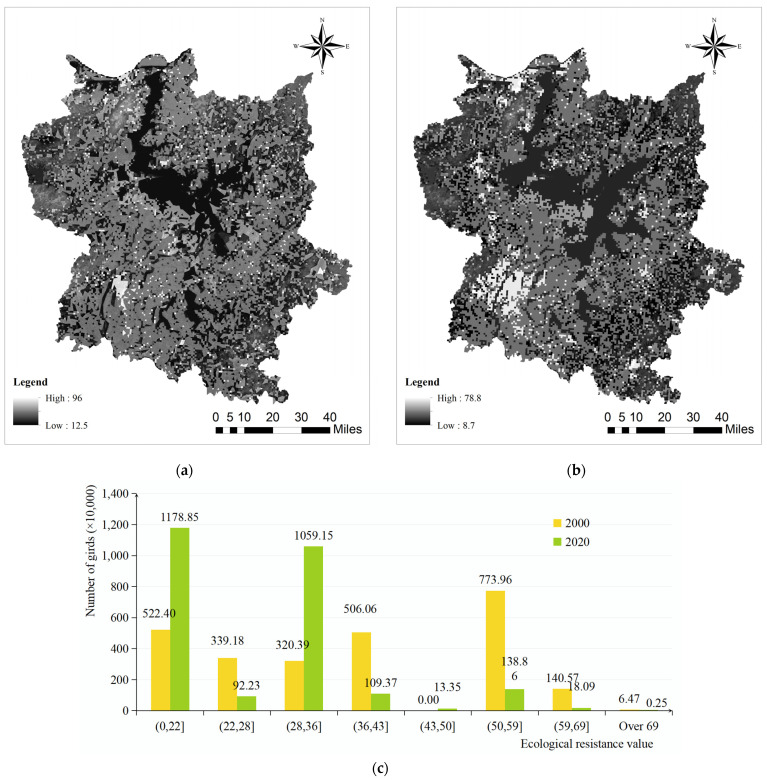
Ecological resistance surface of Poyang Lake Migratory Bird Protect Area: (**a**) 2000; (**b**) 2020; (**c**) ecological resistance pattern of Poyang Lake Migratory Bird Protect Area in 2000 and 2020.

**Figure 5 biology-13-00786-f005:**
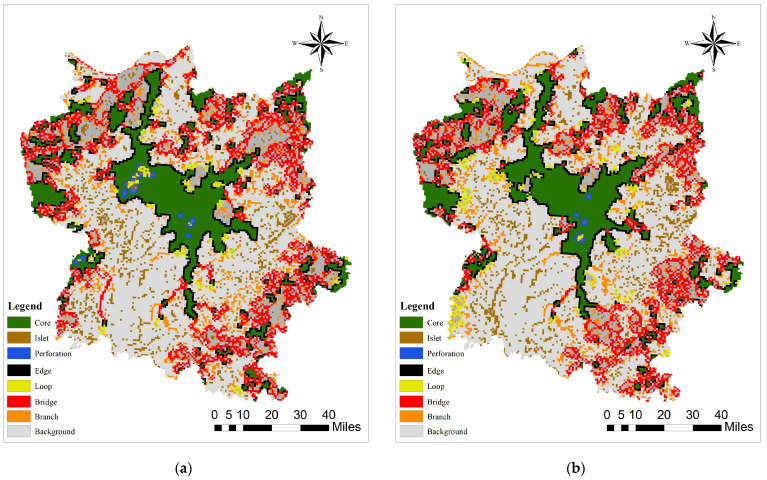
Morphological landscape types of Poyang Lake Migratory Bird Protect Area: (**a**) 2000; (**b**) 2020.

**Figure 6 biology-13-00786-f006:**
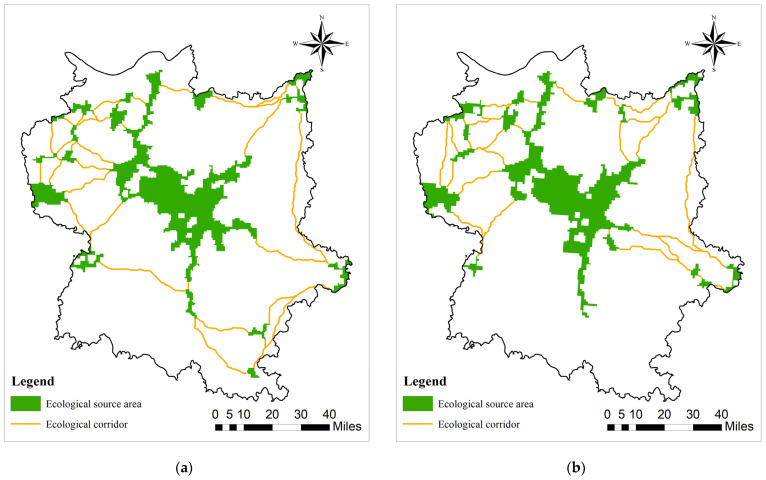
Ecological corridors of Poyang Lake Migratory Bird Protect Area: (**a**) 2000; (**b**) 2020.

**Figure 7 biology-13-00786-f007:**
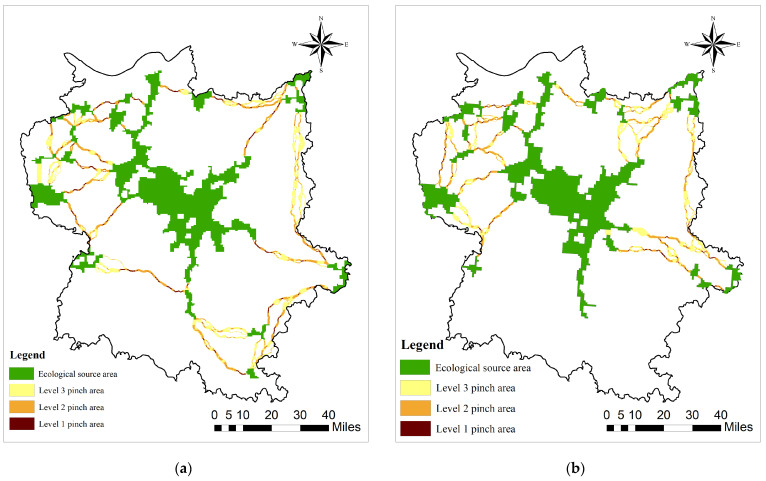
Ecological pinch points of Poyang Lake Migratory Bird Protect Area: (**a**) 2000; (**b**) 2020.

**Figure 8 biology-13-00786-f008:**
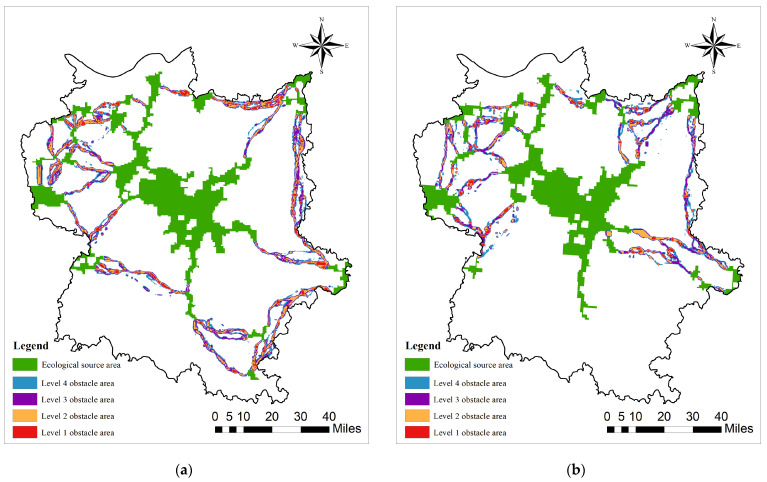
Barrier points of Poyang Lake Migratory Bird Protect Area: (**a**) 2000; (**b**) 2020.

**Table 1 biology-13-00786-t001:** Threat factors and stress intensity.

Threat Factor	Cultivated Land	Urban Land	Rural Settlements	Other Construction Land
Weight	0.15	1	1	0.5
Maximum stress distance/km	1	5	3	3
Decay type	Linear	Exponential	Exponential	Exponential

Note: The data source comes from Natural Capital Project, 2022. InVEST 3.14.2 User’s Guide (https://naturalcapitalproject.stanford.edu/software/invest, accessed on 1 August 2024).

**Table 2 biology-13-00786-t002:** Sensitivity of each habitat type to stressors.

Land Use Type	Habitat Suitability	Threat Factors			
		Cultivated Land	Urban Land	Rural Settlements	Other Construction Land
Paddy field	0.3	0.2	0.8	0.8	0.6
Dry land	0.2	0.3	0.7	0.7	0.5
Forest land	1	0.3	0.7	0.8	0.8
Shrub wood	0.7	0.3	0.8	0.7	0.8
Sparse wood	0.4	0.2	0.7	0.6	0.5
Other forest land	0.6	0.2	0.7	0.6	0.5
High-coverage grassland	0.7	0.3	0.7	0.7	0.6
Medium-coverage grassland	0.5	0.4	0.7	0.7	0.6
Low-coverage grassland	0.3	0.5	0.7	0.7	0.6
River and canals	0.8	0.3	0.8	0.6	0.8
Lakes	0.8	0.3	0.8	0.6	0.8
Reservoirs and ponds	0.7	0.5	0.8	0.5	0.7
Mudflat	0.6	0.6	0.8	0.7	0.7
Urban land	0	0	0	0	0
Rural settlements	0	0	0	0	0
Other construction land	0	0	0	0	0
Marshland	0.5	0.4	0.4	0.2	0.3
Bare land	0	0	0	0	0
Bare rock texture	0	0	0	0	0

Note: The data source comes from Natural Capital Project, 2022. InVEST 3.14.2 User’s Guide (https://naturalcapitalproject.stanford.edu/software/invest, accessed on 1 August 2024).

**Table 3 biology-13-00786-t003:** Ecological resistance assessment indicators.

Resistance Assessment Indicators (Weights)	Grading of Indicators	Landscape Resistance
Land use type (0.5)	Forest	5
	Bush	10
	Grass	20
	Water	20
	Arable land	50
	Bare land	70
	Other land	100
Elevation (0.2)	<50 m	20
	50~100 m	40
	100~500 m	60
	500~1000 m	80
	>1000 m	100
NDVI (0.2)	<0.2	20
	0.2~0.4	40
	0.4~0.6	60
	0.6~0.8	80
	>0.8	100
Slope (0.1)	<5°	20
	5°~10°	40
	10°~20°	60
	20°~30°	80
	>30°	100

Note: The data source comes from Cui et al. (2024) [80] (Cui, W.; Wei, Y.; Su, H.; Liu, X.; Wu, D.; Zhang, N.; Ji, N. Research on Ecological Security Pattern Construction of the Protection and Development Belt of Wuyishan National Park. Research. Of. Environmental. Sciences. 2024, 37, 1–17.).

## Data Availability

Data will be available on request.

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
