# Peer review of "Quantifying the Ecological Performance of Migratory Bird Conservation: Evidence from Poyang Lake Wetlands in China"

_biology, 2024, doi:10.3390/biology13100786_

Round 1

Reviewer 1 Report

Comments and Suggestions for Authors

I am attaching my comments and suggestions related to the work under the title: “Quantifying the Ecological Performance of Migratory Bird Conservation: Evidence from Poyang Lake Wetlands in China”

You do not need a Simple Summary; an Abstract that includes some results from your study should suffice.

L.94: Add references for the InvestModel and GeoSOS model.

L.96: Include a reference for the USLE model, such as:
Wischmeier, W.H. and Smith, D.D. (1978) Predicting Rainfall Erosion Losses. A Guide to Conservation Planning. The USDA Agricultural Handbook No. 537, Maryland.  

L.101: Provide a reference for the Minimum Cumulative Resistance.

L.122-125: Add references for the information you present regarding the number of species. Figure 1 needs to be corrected. Add a map showing a broader view of the study area so that the specific location within China and the coordinates of the lake are clear. It would be beneficial to include the surface area of the lake and the area covered by the Nature Reserves.

A question regarding Figure 1: What is included under "others"? Are these regions, provinces, or cities? Please clarify and rephrase this in the figure and legend.

L.171: Provide a reference for the 1:1,000,000 National Basic Geographic Database.

L.177: Specify which Landsat data you used for both periods—whether a single image or multiple images, and which missions were used (Missions 4, 5, 7, 8?).

L.184: Reference for the SVM method. Specify the resolution of the Land Use Land Cover map that you obtained.

 L.204: Provide a reference for ArcMap 10.8.

L.211-214: A reference is missing for this section of the text. For Tables 1 and 2, provide references if they are based on other authors' work. If the values are original, created by the authors of this paper, please indicate "original."

Also, provide a reference for Table 3.

L.274: Provide a reference for the Linkage Mapper tool.

Figure 6: Change the color scheme used for the map and legend. The current shade of black and white, used for elevation in Figure 1, is not clear. The current color scheme is not easily readable.

The paper is missing a Discussion section, where you should compare your results with previous studies that address the same issue and reference the studies mentioned in the Introduction.

The Conclusion should not be divided into two sub-sections; it should be corrected to remove any additional sub-sections. The Conclusion should be concise, with authors presenting their findings and observations after conducting the research and writing the paper.

Regarding L.222, Land Use data: Did you use machine learning techniques and the Support Vector Machine (SVM) method to obtain the land use/land cover data for the two time periods studied? If so, you should provide an accurate assessment of the classified maps. When using machine learning techniques, there should be polygons or samples used for classification, specifically training samples that teach the machine to classify, as well as validation samples for assessing the accuracy of the results. These should be presented with accuracy metrics such as user, producer, and overall accuracy, along with Kappa statistics. The land use/land cover maps must include these accuracy assessments. If the Kappa statistic is low, the maps may not be reliable, which raises questions about the validity of the other results. This applies if you indeed used machine learning techniques and the SVM method mentioned in your paper.

The paper lacks spatial maps used as input data, such as land use/land cover, NDVI, and slope. These maps should be included either in the paper or in a separate file as an Appendix or Supplementary Material.

Land use/land cover is a critical parameter for understanding migratory bird issues, yet there are no spatial maps for both periods showing land cover categories, nor is there a table or descriptive text discussing the changes observed between the two periods.

The paper requires major revision and changes to be considered for publication.

Author Response

Comments 1: L.94: Add references for the InvestModel and GeoSOS model.

Response 1: Thank you for pointing this out. We agree with this comment. Therefore, we have already added the citation literature of InVEST model and GeoSOS model.

Comments 2: L.96: Include a reference for the USLE model, such as: Wischmeier, W.H. and Smith, D.D. (1978) Predicting Rainfall Erosion Losses. A Guide to Conservation Planning. The USDA Agricultural Handbook No. 537, Maryland.

Response 2: Thank you for pointing this out. We agree with this comment. Therefore, we have added the reference materials of the USLE model in Reference [61].

Comments 3: L.101: Provide a reference for the Minimum Cumulative Resistance.

Response 3: Thank you for pointing this out. We agree with this comment. Therefore, we have added the reference materials of Minimum Cumulative Resistance model.

Comments 4: L.122-125: Add references for the information you present regarding the number of species. Figure 1 needs to be corrected. Add a map showing a broader view of the study area so that the specific location within China and the coordinates of the lake are clear. It would be beneficial to include the surface area of the lake and the area covered by the Nature Reserves. A question regarding Figure 1: What is included under "others"? Are these regions, provinces, or cities? Please clarify and rephrase this in the figure and legend.

Response 4: Thank you for pointing this out. We agree with this comment. Therefore, we have modified the study area map according to the experts' opinions.

Figure 1. Study area.

Comments 5: L.171: Provide a reference for the 1:1,000,000 National Basic Geographic Database.

Response 5: Thank you for pointing this out. We agree with this comment. Therefore, we have added the links of data sources. “The administrative division base map of the sanctuary is sourced from the 1:1,000,000 National Basic Geographic Database (https://www.webmap.cn/).”

Comments 6: L.177: Specify which Landsat data you used for both periods—whether a single image or multiple images, and which missions were used (Missions 4, 5, 7, 8?).

Response 6: Thank you for pointing this out. We agree with this comment. Landsat images consist of multi-band data, and all the original data are sourced from the period between June and September.

Comments 7: L.184: Reference for the SVM method. Specify the resolution of the Land Use Land Cover map that you obtained.

Response 7: Thank you for pointing this out. We agree with this comment. Therefore, we have added the reference materials of the SVM method. The resolution of the Land Use Land Cover map is 1000*1000.

Comments 8: L.204: Provide a reference for ArcMap 10.8.

Response 8: Thank you for pointing this out. We agree with this comment. Therefore, we have added the reference materials of ArcMap 10.8.

Comments 9: L.211-214: A reference is missing for this section of the text. For Tables 1 and 2, provide references if they are based on other authors' work. If the values are original, created by the authors of this paper, please indicate "original." Also, provide a reference for Table 3.

Response 9: Thank you for pointing this out. We agree with this comment. Therefore, we have added citations in L.211-214. And the data source of Figure 1 and Figure 2 come from Natural Capital Project, 2022. InVEST 3.14.2 User’s Guide (https://naturalcapitalproject.stanford.edu/software/invest). The data source of Figure 3 come from Cui et al. (2024) (Cui, W.; Wei, Y.; Su, H.; Liu, X.; Wu, D.; Zhang, N.; Ji, N. Research on Ecological Security Pattern Construction of the Protection and Development Belt of Wuyishan National Park. Research. Of. Environmental. Sciences. 2024, 37, 1-17.).

Comments 10: L.274: Provide a reference for the Linkage Mapper tool.

Response 10: Thank you for pointing this out. We agree with this comment. Therefore, we have added citations.

Comments 11: Figure 6: Change the color scheme used for the map and legend. The current shade of black and white, used for elevation in Figure 1, is not clear. The current color scheme is not easily readable.

Response 11: Thank you for pointing this out. We agree with this comment. Therefore, Figures 1 and 6 have been optimized.

(a)2000

(b)2020

(c)Ecological resistance pattern of Poyang Lake Migratory Bird Protect Area in 2000 and 2020

Figure 4. Ecological resistance surface of Poyang Lake Migratory Bird Protect Area: (a) 2000; (b) 2020; (c) Ecological resistance pattern of Poyang Lake Migratory Bird Protect Area in 2000 and 2020.

Figure 1. Study area.

Comments 12: The paper is missing a Discussion section, where you should compare your results with previous studies that address the same issue and reference the studies mentioned in the Introduction.

Response 12: Thank you for pointing this out. We agree with this comment. Therefore, we have added the discussion part.

5. Discussion

With the continuous intensification of human economic activities, the construction of ecological corridors aimed at optimizing the ecological network system has emerged as a crucial measure to promote human-bird symbiosis within nature reserves and enhance biodiversity. In this paper, we utilized the InVEST model and MCR model in conjunction with circuit theory to construct ecological corridors. We investigated the relationship between ecological connectivity and biodiversity enhancement in the migratory bird reserve of Poyang Lake, revealing the key role of ecological corridors in protecting flagship species.The research in this paper further confirms that human activities consistently disrupt habitat connectivity [27-30], leading to a continuous decline in both biodiversity and habitat quality [2,31-35]. Furthermore, the decrease in ecological habitat connectivity has already become a threat to migratory bird populations [13,36-42]. The ecological network constructed in this paper demonstrates the significance of ecological connectivity in Poyang Lake and emphasizes the necessity of increasing the scale of ecological corridors to protect endangered species and their companion species. The construction of ecological corridors can alleviate, to a certain extent, the contradiction in the human-bird symbiosis in the protected area of Poyang Lake and provides a theoretical basis for regional ecological management. However, despite significant progress in methodology, there are still some deficiencies and areas that need further expansion and improvement. Future studies should consider the disturbance factors of human activities, especially the destruction of ecological corridors by urban expansion, in order to comprehensively assess the sustainability of the ecological network. Additionally, simulating the ecological network of Poyang Lake with climate models will help predict and respond to possible environmental changes.

In summary, this paper highlights the significance of ecological corridors in protecting regional biodiversity through the construction of an ecological network in the Poyang Lake region. It also proposes strategies for optimizing the ecological network to improve the regional ecological environment. However, future research should focus on exploring the dynamic changes of the ecological network and its management strategies using more extensive data and multidisciplinary perspectives. This is essential to achieve a balance between ecological protection and regional development, with a goal of realizing human-bird symbiosis.

Comments 13: Regarding L.222, Land Use data: Did you use machine learning techniques and the Support Vector Machine (SVM) method to obtain the land use/land cover data for the two time periods studied? If so, you should provide an accurate assessment of the classified maps. When using machine learning techniques, there should be polygons or samples used for classification, specifically training samples that teach the machine to classify, as well as validation samples for assessing the accuracy of the results. These should be presented with accuracy metrics such as user, producer, and overall accuracy, along with Kappa statistics. The land use/land cover maps must include these accuracy assessments. If the Kappa statistic is low, the maps may not be reliable, which raises questions about the validity of the other results. This applies if you indeed used machine learning techniques and the SVM method mentioned in your paper.

Response 13: Thank you for pointing this out. We agree with this comment. The Kappa statistics of the LUCC data produced based on machine learning technology and the SVM method are 0.834 and 0.892, which are relatively high values and have reliability.

Comments 14: The paper lacks spatial maps used as input data, such as land use/land cover, NDVI, and slope. These maps should be included either in the paper or in a separate file as an Appendix or Supplementary Material.

Response 14: Thank you for pointing this out. We agree with this comment. The above figures are supplemented in the appendix.

Appendix

(a)2000

(b)2020

Figure i. Land use/cover change pattern of Poyang Lake Migratory Bird Protect Area: (a) 2000; (b) 2020.

(a)2000

(b)2020

Figure ii. NDVI pattern of Poyang Lake Migratory Bird Protect Area: (a) 2000; (b) 2020.

Figure iii. Slope pattern of Poyang Lake Migratory Bird Protect Area.

Reviewer 2 Report

Comments and Suggestions for Authors

The study on quantifying the ecological performance of migratory bird conservation based on evidence from Poyang Lake wetlands in China adds to existing knowledge and has the potential for publication in Biology. However, several aspects of the manuscript require improvement and clarification before a positive decision can be reached.

Why is the discussion section missing? The discussion is essential because it interprets the findings, places them within the context of existing literature, and highlights the implications of the results. It also addresses potential limitations and suggests directions for future research, helping to frame the study’s contributions to the field.

What are the potential limitations of this study, particularly concerning the methods used in comparison to those in other similar studies? Identifying these limitations can provide insights into the reliability and validity of the findings, enhancing the overall understanding of the study's contributions to the field.

L165 The map of the study area requires minor improvements. Specifically, the unit of measurement for the elevation scale (labeled 1 to 5) needs to be clearly indicated. Additionally, the current color gradient for elevation, where white represents low values and black represents high values, is problematic because white is also used to denote “Other reserves.” A change is needed to enhance clarity and accurately reflect the topography and distribution of reserves.

L281‒L283 states that the habitat quality of the Poyang Lake Migratory Bird Protection Area improved from 2000 to 2020, likely due to effective conservation efforts or favorable environmental changes. A statistical comparison between the years 2000 and 2020 based on the data presented in Fig. 3 would be valuable for quantifying the extent of this habitat quality improvement. Figures can be organized by including Fig. 3 and these new results in Fig. 2.

L341 A statistical comparison between the years 2000 and 2020 based on the data presented in Fig. 5 would be valuable for quantifying the extent of this habitat degradation. Figures can be organized by including Fig. 5 and these new results in Fig. 4.

L374 A statistical comparison between the years 2000 and 2020 based on the data presented in Fig. 7 would be valuable for quantifying the extent of this ecological resistance. Figures can be organized by including Fig. 7 and these new results in Fig. 6.

Comments on the Quality of English Language

Moderate editing of English language required.

Author Response

Comments 1: Why is the discussion section missing? The discussion is essential because it interprets the findings, places them within the context of existing literature, and highlights the implications of the results. It also addresses potential limitations and suggests directions for future research, helping to frame the study’s contributions to the field. What are the potential limitations of this study, particularly concerning the methods used in comparison to those in other similar studies? Identifying these limitations can provide insights into the reliability and validity of the findings, enhancing the overall understanding of the study's contributions to the field.

Response 1: Thank you for pointing this out. We agree with this comment. Therefore, discussion has been added and limitations analyzed.

5. Discussion

With the continuous intensification of human economic activities, the construction of ecological corridors aimed at optimizing the ecological network system has emerged as a crucial measure to promote human-bird symbiosis within nature reserves and enhance biodiversity. In this paper, we utilized the InVEST model and MCR model in conjunction with circuit theory to construct ecological corridors. We investigated the relationship between ecological connectivity and biodiversity enhancement in the migratory bird reserve of Poyang Lake, revealing the key role of ecological corridors in protecting flagship species.The research in this paper further confirms that human activities consistently disrupt habitat connectivity [27-30], leading to a continuous decline in both biodiversity and habitat quality [2,31-35]. Furthermore, the decrease in ecological habitat connectivity has already become a threat to migratory bird populations [13,36-42]. The ecological network constructed in this paper demonstrates the significance of ecological connectivity in Poyang Lake and emphasizes the necessity of increasing the scale of ecological corridors to protect endangered species and their companion species. The construction of ecological corridors can alleviate, to a certain extent, the contradiction in the human-bird symbiosis in the protected area of Poyang Lake and provides a theoretical basis for regional ecological management. However, despite significant progress in methodology, there are still some deficiencies and areas that need further expansion and improvement. Future studies should consider the disturbance factors of human activities, especially the destruction of ecological corridors by urban expansion, in order to comprehensively assess the sustainability of the ecological network. Additionally, simulating the ecological network of Poyang Lake with climate models will help predict and respond to possible environmental changes.

In summary, this paper highlights the significance of ecological corridors in protecting regional biodiversity through the construction of an ecological network in the Poyang Lake region. It also proposes strategies for optimizing the ecological network to improve the regional ecological environment. However, future research should focus on exploring the dynamic changes of the ecological network and its management strategies using more extensive data and multidisciplinary perspectives. This is essential to achieve a balance between ecological protection and regional development, with a goal of realizing human-bird symbiosis.

Comments 2: L165 The map of the study area requires minor improvements. Specifically, the unit of measurement for the elevation scale (labeled 1 to 5) needs to be clearly indicated. Additionally, the current color gradient for elevation, where white represents low values and black represents high values, is problematic because white is also used to denote “Other reserves.” A change is needed to enhance clarity and accurately reflect the topography and distribution of reserves.

Response 2: Thank you for pointing this out. We agree with this comment. Therefore, Figure 1 and its legend have been corrected.

Figure 1. Study area.

Comments 3: L281‒L283 states that the habitat quality of the Poyang Lake Migratory Bird Protection Area improved from 2000 to 2020, likely due to effective conservation efforts or favorable environmental changes. A statistical comparison between the years 2000 and 2020 based on the data presented in Fig. 3 would be valuable for quantifying the extent of this habitat quality improvement. Figures can be organized by including Fig. 3 and these new results in Fig. 2.

Response 3: Thank you for pointing this out. We agree with this comment. Therefore, we have changed.

(a)2000

(b)2020

(c)Habitat quality pattern of Poyang Lake Migratory Bird Protect Area in 2000 and 2020

Figure 2. Habitat Quality of Poyang Lake Migratory Bird Protect Area: (a) 2000; (b) 2020; (c)Habitat quality pattern of Poyang Lake Migratory Bird Protect Area in 2000 and 2020.

Comments 4: L341 A statistical comparison between the years 2000 and 2020 based on the data presented in Fig. 5 would be valuable for quantifying the extent of this habitat degradation. Figures can be organized by including Fig. 5 and these new results in Fig. 4.

Response 4: Thank you for pointing this out. We agree with this comment. Therefore, we have changed.

(a)2000

(b)2020

(c)Habitat quality pattern of Poyang Lake Migratory Bird Protect Area in 2000 and 2020

Figure 3. Degree of habitat degradation at the Poyang Lake Migratory Bird Protect Area: (a) 2000; (b) 2020; (c)Habitat quality pattern of Poyang Lake Migratory Bird Protect Area in 2000 and 2020.

Comments 5: L374 A statistical comparison between the years 2000 and 2020 based on the data presented in Fig. 7 would be valuable for quantifying the extent of this ecological resistance. Figures can be organized by including Fig. 7 and these new results in Fig. 6.

Response 5: Thank you for pointing this out. We agree with this comment. Therefore, we have changed.

(a)2000

(b)2020

(c)Ecological resistance pattern of Poyang Lake Migratory Bird Protect Area in 2000 and 2020

Figure 4. Ecological resistance surface of Poyang Lake Migratory Bird Protect Area: (a) 2000; (b) 2020; (c) Ecological resistance pattern of Poyang Lake Migratory Bird Protect Area in 2000 and 2020.

Reviewer 3 Report

Comments and Suggestions for Authors

In the paper “Quantifying the Ecological Performance of Migratory Bird Conservation: Evidence from Poyang Lake Wetlands in China” the authors investigates the spatiotemporal dynamics of habitat quality in the Poyang Lake migratory bird protection area, employing the InVEST model, Minimum Cumulative Resistance model, and circuit theory to elucidate the spatial patterns of land use for migratory bird habitats at Poyang Lake. This manuscript is well organized, and the drawn conclusions are coherent with the obtained results. Despite I have enjoyed reading your paper; I feel that it needs to be corrected by a native English speaker because I have seen a few grammatical errors. I hope to provide very useful suggestions to improve the overall clarity of your study as well as the quality of your analysis. I think that my suggestions look feasible to you, and I believe you will be able to address them. Thus, please take care to do a full revision of your manuscript according to all my comments. Improvements based on my comments will be crucial for acceptance. I have some concerns and suggestions for each aspect of the manuscript. Please see below:

Lines 29 – 35: To give more emphasis to your results.

Lines 36 – 37: To arrange the keywords alphabetically.

Lines 45 – 56: To reduce this part of the manuscript.

Lines 57 – 58: I think that you should add more recent references as examples to support your sentence: “Previous studies have made significant advancements in the conservation of habitats within protected natural areas.”. I would like to suggest:

Bosso, L., et al., (2024). Integrating citizen science and spatial ecology to inform management and conservation of the Italian seahorses. Ecological Informatics, 79, 102402.

Rego, M. A., Del‐Rio, G., & Brumfield, R. T. (2024). Subspecies‐level distribution maps for birds of the Amazon basin and adjacent areas. Journal of Biogeography, 51(1), 14-28.

Lines 84 – 86: I think that this sentence should be moved between the lines 105 – 114.

Lines 105 – 114: Please explain in detail your hypothesis and predictions. You need to expand this section if you would want to express exactly what you want to do.

Lines 142 – 144: I think that you should add more recent references as examples to support your sentence: “However, this growth has led to a paradoxical situation where the available habitat area fails to keep pace with the expanding population due to human disturbances and encroachment.”. I would like to suggest:

Fraissinet, M., et al.,(2023). Responses of avian assemblages to spatiotemporal landscape dynamics in urban ecosystems. Landscape Ecology, 38(1), 293-305.

Purwanto, Andradi‐Brown, D. A., et al., (2021). The Bird's Head Seascape Marine Protected Area network—Preventing biodiversity and ecosystem service loss amidst rapid change in Papua, Indonesia. Conservation Science and Practice, 3(6), e393.

Line 165: Please add also a map of the world showing your study area.

Lines 176 – 188: To add a link to the database where you have downloaded/stored the data.

Lines 204 – 209: Please describe in details all the ArcGIS and InVEST processes used in your analyses.

Line 275: Please describe in details all setting used in your Circuit analysis.

Lines 295 – 397: The figures 2, 4, 6 and 8 must be included in only one figure to make a panel. The figures 3, 5 and 7 must be included in only one figure to make a panel.

Lines 430: Where is the discussion section? You cannot send a paper in review without a discussion section where you discuss your results. Please add this section in your manuscript.

Comments on the Quality of English Language

Extensive editing of English language required.

Author Response

Comments 1: Lines 29 – 35: To give more emphasis to your results.

Response 1: Thank you for pointing this out. We agree with this comment. Therefore, we have changed.

The research results in this paper are as follows. (i) During the period of 2000 to 2020, there was an overall decline in habitat quality within the study area, indicating a clear trend of habitat degradation. However, it is worth noting that there was an increase in habitat quality in certain local areas within the protected area. (ii) The ecological resistance values in the core area of the migratory bird reserve in Poyang Lake are generally low. However, the ecological resistance values of the habitats have shown a consistent increase from 2000 to 2020. Additionally, there has been a significant decrease in the density of ecological corridors during this time period. (iii) Over the period from 2000 to 2020, both the number and connectivity of ecological corridors decreased while their integrity and functionality degraded. Consequently, this weakened role of the ecological network has had implications on maintaining regional biodiversity and ecosystem service functions.

Comments 2: Lines 36 – 37: To arrange the keywords alphabetically.

Response 2: Thank you for pointing this out. We agree with this comment. Therefore, we have changed. Adjusted to list keywords in alphabetical order.

Keywords: avian conservation; coexistence of humans and birds; ecological modeling; ecological network; GIS; habitat quality; Poyang Lake

Comments 3: Lines 45 – 56: To reduce this part of the manuscript.

Response 3: Thank you for pointing this out. We agree with this comment. Therefore, we have changed.

The climate crisis is exacerbating floods and drought disasters [8-11], leading to severe degradation or loss of traditional habitats for species [12]. Habitat fragmentation, disruption of waterway connections, and disappearance of traditional foraging grounds are further worsening the ecological crisis [13]. It is crucial to take urgent measures to enhance the connectivity of wetland ecosystems in order to maintain biodiversity, as it holds significant ecological significance [14-16]. Hence, the investigation of ecological connectivity is essential to ensure the habitat security of wetland migratory bird sanctuaries.

Comments 4: Lines 57 – 58: I think that you should add more recent references as examples to support your sentence: “Previous studies have made significant advancements in the conservation of habitats within protected natural areas.”. I would like to suggest: Bosso, L., et al., (2024). Integrating citizen science and spatial ecology to inform management and conservation of the Italian seahorses. Ecological Informatics, 79, 102402. Rego, M. A., Del‐Rio, G., & Brumfield, R. T. (2024). Subspecies‐level distribution maps for birds of the Amazon basin and adjacent areas. Journal of Biogeography, 51(1), 14-28.

Response 4: Thank you for pointing this out. We agree with this comment. Therefore, we have added new references.

Comments 5: Lines 84 – 86: I think that this sentence should be moved between the lines 105 – 114.

Response 5: Thank you for pointing this out. We agree with this comment. Therefore, it has been moved as you suggested.

Comments 6: Lines 105 – 114: Please explain in detail your hypothesis and predictions. You need to expand this section if you would want to express exactly what you want to do.

Response 6: Thank you for pointing this out. We agree with this comment. Therefore, we have changed.

This paper consists of five sections. The second section provides an overview of the study area, data sources, and research methods. The third section examines the changes in habitat quality in the study area from 2000 to 2020, develops an ecological resistance surface, identifies ecological corridors, and establishes an ecological network for analyzing the pattern of ecological change. The fourth section presents the research conclusion and policy implications. The fifth section includes further discussion on the topic.

Comments 7: Lines 142 – 144: I think that you should add more recent references as examples to support your sentence: “However, this growth has led to a paradoxical situation where the available habitat area fails to keep pace with the expanding population due to human disturbances and encroachment.”. I would like to suggest: Fraissinet, M., et al.,(2023). Responses of avian assemblages to spatiotemporal landscape dynamics in urban ecosystems. Landscape Ecology, 38(1), 293-305. Purwanto, Andradi‐Brown, D. A., et al., (2021). The Bird's Head Seascape Marine Protected Area network—Preventing biodiversity and ecosystem service loss amidst rapid change in Papua, Indonesia. Conservation Science and Practice, 3(6), e393.

Response 7: Thank you for pointing this out. We agree with this comment. Therefore, references have been added.

Comments 8: Line 165: Please add also a map of the world showing your study area.

Response 8: Thank you for pointing this out. We agree with this comment. Therefore, Figures 1 has been reorganized.

Figure 1. Study area.

Comments 9: Lines 176 – 188: To add a link to the database where you have downloaded/stored the data.

Response 9: Thank you for pointing this out. We agree with this comment. Therefore, we have added the data link.

Comments 10: Lines 204 – 209: Please describe in details all the ArcGIS and InVEST processes used in your analyses.

Response 10: Thank you for pointing this out. We agree with this comment. Therefore, the invest calculation formula has been supplemented, and the ArcGIS process is mainly presented in picture form.

Comments 11: Please describe in details all setting used in your Circuit analysis.

Response 11: Thank you for pointing this out. We agree with this comment. Therefore, we have changed. Circuit theory is a theory that is only used to explain the principles of the formation of ecological corridors in this study. "The model takes advantage of the random walk properties of electric charges and combines circuit theory with motion ecology. In this model, the ecological landscape is regarded as a conductive surface, the organisms are regarded as mobile electrons, and the movement of organisms is regarded as an electric current. Based on this, a random walk model is constructed to obtain the path with the least cost of species migration, namely the ecological corridor." Therefore, there is no specific setting of circuit theory. linkage mapper module of Arcmap10.8 software implements the ecological corridor.

2.3.3. Construction of Ecological Network Based on Circuit Theory

· Circuit Theory.

The concept of circuit theory was advanced by McRae et al. (2008) [65], which models the interconnections among diverse ecological landscapes by capitalizing on the random walk characteristic of charges. This model fuses circuit theory with movement ecology by considering ecological landscapes as conductive surfaces and organisms as mobile electrons. The movement of organisms is regarded as current, and a random walk model is erected on this foundation to acquire the path with the least energy expenditure for species migration, namely, ecological corridors.

· Extraction of ecological corridor.

Ecological corridors serve as significant conduits for the material cycling, gene flow, and information transmission within ecosystems, exerting a vital role in connecting various ecological source areas [87-89]. The construction of ecological corridors presented in this paper is founded on the circuit theory [89,90], taking ecological source areas and ecological resistance surfaces as input elements and being implemented through the Linkage Mapper tool module of ArcMap 10.8 software. The connection degree model grounded on the circuit theory considers the traits of species' random migration and is capable of accurately simulating the actual circumstances of species migration [85], thereby constructing an ecological network for the protection of migratory birds in Poyang Lake.

· Identification of Ecological Pinch Point and Ecological Barrier Point.

Ecological pinch points represent crucial junctions that ensure the connectivity among sources and exert an irreplaceable role in maintaining the integrity of ecosystems [91]. Due to the irreplaceable characteristic of ecological pinch points, organisms are prone to select this area during their migration. Consequently, any damage to this area would exert a fatal influence on the integrity of the ecosystem and biodiversity. Hence, the ecological pinch points area constitutes a key protected area within the entire study region. Utilizing the Pinchpoint Mapper tool within the Linkage Mapper module, which integrates minimum cost path and circuit theory in the study area, initially employing Circuitscape software to identify areas with high current density. Subsequently, Pinchpoint Mapper is utilized to eliminate areas that contribute minimally to regional connectivity, thus identifying the path of least resistance and critical pinch points. This ultimately leads to the identification of key node areas within the ecological corridor of the study area. This paper identifies ecological pinch points in the Poyang Lake Wetland Bird Reserve using the raster centrality option in the All-to-one pattern recognition model. The All-to-one pattern establishes a connection between a core area and the ground, and then distributes current into the remaining cores by iterating through all core areas. Based on the iterative results, the ecological pinch points in the Poyang Lake Wetland Bird Reserve are determined.

Ecological barrier points refer to areas that hinder the quality of ecological corridors between source areas [29]. The occurrence of barrier points is influenced by multiple factors, and is typically more significantly affected by human interference. Based on the land use types in the study area, distinct solutions are adopted for various problems to minimize the resistance value in the region and enhance the connectivity among landscapes as much as possible through ecological restoration. In this study, Barrier Mapper tool within Linkage Mapper module is employed to identify the highly obstructive areas within the ecological corridor. The minimum search radius is set at 300m, the maximum search radius at 900m, and the step size at 300m. The obstacle points of the study area were searched through the moving window approach. The operation of the Barrier Mapper tool is based on the Linkage Pathways Tool, which can detect both complete barriers that completely impede species migration and barrier areas that have some degree of impeding effect but are not completely impeding species migration [92]. In this study, only barrier areas that completely impede species migration are considered to determine the ecological barrier point regions in the Poyang Lake bird protection area.

Comments 12: Lines 295 – 397: The figures 2, 4, 6 and 8 must be included in only one figure to make a panel. The figures 3, 5 and 7 must be included in only one figure to make a panel.

Response 12: Thank you for pointing this out. We agree with this comment. Therefore, we have changed.

(a)2000

(b)2020

(c)Habitat quality pattern of Poyang Lake Migratory Bird Protect Area in 2000 and 2020

Figure 2. Habitat Quality of Poyang Lake Migratory Bird Protect Area: (a) 2000; (b) 2020; (c)Habitat quality pattern of Poyang Lake Migratory Bird Protect Area in 2000 and 2020.

(a)2000

(b)2020

(c)Habitat quality pattern of Poyang Lake Migratory Bird Protect Area in 2000 and 2020

Figure 3. Degree of habitat degradation at the Poyang Lake Migratory Bird Protect Area: (a) 2000; (b) 2020; (c)Habitat quality pattern of Poyang Lake Migratory Bird Protect Area in 2000 and 2020.

(a)2000

(b)2020

(c)Ecological resistance pattern of Poyang Lake Migratory Bird Protect Area in 2000 and 2020

Figure 4. Ecological resistance surface of Poyang Lake Migratory Bird Protect Area: (a) 2000; (b) 2020; (c) Ecological resistance pattern of Poyang Lake Migratory Bird Protect Area in 2000 and 2020.

(a)2000

(b)2020

Figure 5. Morphological landscape type of of Poyang Lake Migratory Bird Protect Area: (a) 2000; (b) 2020.

(a)2000

(b)2020

Figure 6. Ecological corridors of Poyang Lake Migratory Bird Protect Area: (a) 2000; (b) 2020.

(a)2000

(b)2020

Figure 7. Ecological pinch points of Poyang Lake Migratory Bird Protect Area: (a) 2000; (b) 2020.

(a)2000

(b)2020

Figure 8. Barrier points of Poyang Lake Migratory Bird Protect Area: (a) 2000; (b) 2020.

Comments 13: Lines 430: Where is the discussion section? You cannot send a paper in review without a discussion section where you discuss your results. Please add this section in your manuscript.

Response 13: Thank you for pointing this out. We agree with this comment. Therefore, we have added the part of discussion.

5. Discussion

With the continuous intensification of human economic activities, the construction of ecological corridors aimed at optimizing the ecological network system has emerged as a crucial measure to promote human-bird symbiosis within nature reserves and enhance biodiversity. In this paper, we utilized the InVEST model and MCR model in conjunction with circuit theory to construct ecological corridors. We investigated the relationship between ecological connectivity and biodiversity enhancement in the migratory bird reserve of Poyang Lake, revealing the key role of ecological corridors in protecting flagship species.The research in this paper further confirms that human activities consistently disrupt habitat connectivity [27-30], leading to a continuous decline in both biodiversity and habitat quality [2,31-35]. Furthermore, the decrease in ecological habitat connectivity has already become a threat to migratory bird populations [13,36-42]. The ecological network constructed in this paper demonstrates the significance of ecological connectivity in Poyang Lake and emphasizes the necessity of increasing the scale of ecological corridors to protect endangered species and their companion species. The construction of ecological corridors can alleviate, to a certain extent, the contradiction in the human-bird symbiosis in the protected area of Poyang Lake and provides a theoretical basis for regional ecological management. However, despite significant progress in methodology, there are still some deficiencies and areas that need further expansion and improvement. Future studies should consider the disturbance factors of human activities, especially the destruction of ecological corridors by urban expansion, in order to comprehensively assess the sustainability of the ecological network. Additionally, simulating the ecological network of Poyang Lake with climate models will help predict and respond to possible environmental changes.

In summary, this paper highlights the significance of ecological corridors in protecting regional biodiversity through the construction of an ecological network in the Poyang Lake region. It also proposes strategies for optimizing the ecological network to improve the regional ecological environment. However, future research should focus on exploring the dynamic changes of the ecological network and its management strategies using more extensive data and multidisciplinary perspectives. This is essential to achieve a balance between ecological protection and regional development, with a goal of realizing human-bird symbiosis.

Comments 14: Extensive editing of English language required.

Response 14: Thank you for pointing this out. We agree with this comment. Therefore, we improved the English.

Round 2

Reviewer 1 Report

Comments and Suggestions for Authors

The paper's authors corrected it according to the instructions and suggestions of the reviewer, but the paper still has some errors. The manuscript can be accepted when the authors correct the chapters. The Conclusion chapter cannot come before the Discussion. The Conclusion is at the end of the manuscript.

Author Response

We are grateful to the editor and reviewers for their positive and constructive comments and criticisms concerning our manuscript, “Quantifying the Ecological Performance of Migratory Bird Conservation: Evidence from Poyang Lake Wetlands in China” (ID: biology-3149310). These comments and criticisms are very helpful for revising and improving our paper. We have made necessary corrections and changes in response to them. We hope that you will find our revised manuscript is acceptable for publication. Of course, we will make additionally changes if there remain unaddressed or inadequately addressed comments.

Comments 1: The paper's authors corrected it according to the instructions and suggestions of the reviewer, but the paper still has some errors. The manuscript can be accepted when the authors correct the chapters. The Conclusion chapter cannot come before the Discussion. The Conclusion is at the end of the manuscript.

Response 1: Thank you for pointing this out. We agree with this comment. Therefore, the discussion section has been moved to precede the conclusion section.

Reviewer 2 Report

Comments and Suggestions for Authors

The authors have thoroughly addressed previous comments, leading to significant improvements in the manuscript. The revised version exhibits greater clarity, and the findings are of substantial interest to the scientific community. However, several aspects still require attention.

Text within the figures should be removed, except for the figure number, as it is already provided in the caption.

Double-check the language quality to avoid duplicated information and ensure clarity throughout the manuscript.

L637 In standard papers, the Discussion section is typically placed before the Conclusions. Please verify the consistency of the manuscript structure with the Guide for Authors in Biology.

Comments on the Quality of English Language

Minor editing of English language required.

Author Response

We are grateful to the editor and reviewers for their positive and constructive comments and criticisms concerning our manuscript, “Quantifying the Ecological Performance of Migratory Bird Conservation: Evidence from Poyang Lake Wetlands in China” (ID: biology-3149310). These comments and criticisms are very helpful for revising and improving our paper. We have made necessary corrections and changes in response to them. We hope that you will find our revised manuscript is acceptable for publication. Of course, we will make additionally changes if there remain unaddressed or inadequately addressed comments.

Comments 1: Text within the figures should be removed, except for the figure number, as it is already provided in the caption.

Response 1: Thank you for pointing this out. We agree with this comment. Therefore, the titles under the series of figures in the Figure have been removed to avoid duplicating the main title of the Figure.

Comments 2: Double-check the language quality to avoid duplicated information and ensure clarity throughout the manuscript.

Response 2: Thank you for pointing this out. We agree with this comment. Therefore, we carefully reviewed the language quality and made polishing adjustments to the language expression.

Comments 3: L637 In standard papers, the Discussion section is typically placed before the Conclusions. Please verify the consistency of the manuscript structure with the Guide for Authors in Biology.

Response 3: Thank you for pointing this out. We agree with this comment. Therefore, the discussion section has been moved to precede the conclusion section.

Reviewer 3 Report

Comments and Suggestions for Authors

Well done!

Comments on the Quality of English Language

Well done!

Author Response

We are grateful to the editor and reviewers for their positive and constructive comments and criticisms concerning our manuscript, “Quantifying the Ecological Performance of Migratory Bird Conservation: Evidence from Poyang Lake Wetlands in China” (ID: biology-3149310). These comments and criticisms are very helpful for revising and improving our paper. We have made necessary corrections and changes in response to them. We hope that you will find our revised manuscript is acceptable for publication. Of course, we will make additionally changes if there remain unaddressed or inadequately addressed comments.

Comments 1: Minor editing of English language required.

Response 1: Thank you for pointing this out. We agree with this comment. Therefore, we carefully reviewed the language quality and made polishing adjustments to the language expression.
